# COVID-19 Vaccine Mandates for Healthcare Professionals in the United States

**DOI:** 10.3390/vaccines10091425

**Published:** 2022-08-30

**Authors:** Rohan Rao, Abigail Koehler, Katrina Beckett, Soma Sengupta

**Affiliations:** 1Department of Neurology and Rehabilitation Medicine, University of Cincinnati College of Medicine, Cincinnati, OH 45267, USA; 2Department of Health Administration, University of Cincinnati, Cincinnati, OH 45221, USA

**Keywords:** COVID-19, corona virus, vaccines, vaccine mandates, pandemic, community health, public health, healthcare workers

## Abstract

Healthcare workers (HCWs) need to be vaccinated against COVID-19 because they care for vulnerable patients. Hesitation to receiving the COVID-19 vaccine stems from the argument of bodily autonomy, novel mRNA vaccine technology, and conspiracy theories. However, vaccinations may prevent thousands of hospitalizations and deaths. HCWs have previously complied with other required vaccinations to care for children, elderly, and immunocompromised patients. Yet, COVID-19 vaccination mandates in the healthcare setting have been faced with resistance and subsequent staffing shortages. As HCWs display their hesitation to the vaccine, the community loses trust in its efficacy and safety. Speculation on pharmaceutical profiteering has also contributed to vaccine mistrust. As the pandemic continues, the healthcare field must decide on a course of action: adhere to vaccination mandates and cope with decreased staffing, repeal vaccination mandates to recover staff, rely on personal protective equipment (PPE) alone for protection, or do nothing and expect survival through herd immunity. To date, the United States has chosen to mandate COVID-19 vaccinations for any healthcare worker employed by Medicare and/or Medicaid-accepting facilities, allowing allergy and religious exemptions. This COVID-19 vaccination mandate for HCWs ethically protects the vulnerable people who HCWs vow to care for.

## 1. Introduction

Many healthcare workers (HCWs) disagree with mandatory vaccination against COVID-19 since they feel this is an infringement of their autonomy [1,2]. However, HCWs take care of vulnerable populations. Vaccination serves as a valuable tool to protect HCWs from contracting and spreading serious illnesses while serving these vulnerable populations [3]. As Vilches et al. found, “A 50% increase in daily vaccine doses administered to previously unvaccinated individuals is projected to prevent 30,727 hospitalizations and 11,937 deaths” [3].

A systematic review of HCWs globally showed that of all Americans infected with COVID-19, 19% were HCWs [4]. Those working in Internal and Family Medicine had the highest rate of death, closely followed by mental health nursing professionals. Demographic data suggest that most HCWs infected in the U.S. were white (2743) compared with other ethnicities (1058). HCWs’ infection, mortality, and transmission rates have shifted since vaccinations became mandated for HCWs. Amongst U.S. HCWs, there is up to a 96.3% vaccine effectiveness when both doses are received [5].

Before the pandemic, HCWs were required to be vaccinated against many transmissible diseases to take care of vulnerable populations, including children, the elderly, and immunocompromised people. COVID-19 vaccine hesitancy is attributable to many elements, including the novelty of the mRNA vaccine technology, conspiracy theories, and politicization of the choice [6,7]. A meta-analysis showed that 22.51% out of 76,471 HCWs worldwide reported COVID-19 vaccine hesitancy [8]. This is not a new phenomenon as another meta-analysis showed that 39% of HCWs worldwide disagree with mandatory flu vaccination [9].

The implementation of vaccine mandates for healthcare professionals has led to severe shortages of HCWs in many different capacities [10,11]. According to Whelan and Evans, who cite data from the Centers for Disease Control and Prevention (CDC), 30% of HCWs were unvaccinated in September 2021 [10]. Vaccine hesitancy in the healthcare community has had a breakthrough effect on patients. For example, numerous potentially preventable deaths in nursing homes from the delta variant of COVID-19 were caused by HCWs not having been vaccinated [12]. 

According to the Supreme Court ruling, the COVID-19 vaccine is federally mandated for all HCWs where facilities take Medicare and Medicaid [13]. The deadline for HCWs to be fully vaccinated was 15 March 2022. Bioethicists argue that despite the apprehension that HCWs might have regarding these vaccines, they are ethically obliged to take care of vulnerable patient populations [14]. Thus, they must ensure that they become vaccinated. This white paper provides support for the COVID-19 vaccine mandate for U.S. healthcare professionals while exploring the alternatives [15]. 

## 2. Problem Statement

The COVID-19 vaccine mandate for HCWs is essential, as many HCWs are subjected to morbidity, resulting in long-term psychological scarring and mortality [16]. On 16 July 2020, over 100,000 COVID-19 cases were reported by healthcare professionals with 641 deaths [17]. As of 10 February 2022, there had been 76,448,067 confirmed cases of COVID-19 in the US, and 902,189 confirmed deaths from COVID-19 [18]. As Director-General Tedros Adhanom Ghebreyesus stated, “The pandemic is a powerful demonstration of just how much we rely on health workers and how vulnerable we all are when the people who protect our health are themselves unprotected” [19].

In the U.S., Operation Warp Speed was a public-private partnership instituted on 15 May 2020. This was created to enable COVID-19 vaccine research and subsequent vaccine distribution by January 2021 [20]. However, uptake of the vaccines continues to be low, even for HCWs. One study notes that herd immunity could occur if 55–82% of the population were vaccinated, accounting for environmental and biological variables [21]. These authors discuss vaccine hesitancy and note that during Operation Warp Speed, only 30% of those surveyed would agree to be vaccinated, failing to meet their herd immunity threshold. Some epidemiologists make the argument that in a “novel” virus, herd immunity may not even be possible. Regardless, vaccination significantly reduces hospitalizations and deaths from COVID-19, emphasizing the utility of the vaccine [22].

Following the Supreme Court ruling, the Center for Medicare and Medicaid Services mandated that HCWs be vaccinated by 15 March 2022, in all states, apart from Texas, where a preliminary injunction prevents this requirement [13]. The U.S. mandate was preceded by Italy, which became the first European country to mandate the COVID-19 vaccination for healthcare workers in April 2021 [23,24,25]. Despite the European precedent, U.S. HCWs have resisted this vaccine mandate, causing severe staff shortages in medical settings [1]. Becker’s Hospital Review created a list of employee departures at 55 hospitals that was last updated February 2022 [11]. This report details thousands of employee resignations or terminations based on vaccine hesitancy. For example, a total of 1400 employees at N.Y.-based Northwell Health either resigned or were terminated for refusing the COVID-19 vaccine. 

Further complicating the landscape surrounding vaccine requirements is the politicization of both sides [2]. As the pandemic progressed, an individual’s vaccination preference became a partisan choice rather than a personal health choice. Suddenly, a vaccine mandate could impinge not only on one’s healthcare autonomy but also one’s political beliefs [2,26].

Attempting to navigate this complexity, the Supreme Court ruling contains the following two statements, which are the underpinnings of the current vaccine mandate for healthcare professionals: “[The medical professional’s] core mission is to ensure that the healthcare providers who care for Medicare and Medicaid patients protect their patients’ health and safety” and “ensuring that providers take steps to avoid transmitting a dangerous virus to their patients is consistent with the fundamental principle of the medical profession: First, do no harm.”

## 3. Considering Key Stakeholders

The key stakeholders in this issue are the HCWs, patients, and pharmaceutical companies responsible for the rollout of the vaccines.

Regarding HCWs, it is critical that each healthcare worker is an “ambassador” for patients [14]. Physician vaccination has been identified as critical in promoting patient trust in the COVID-19 vaccine [27,28]. This is not a new phenomenon—physician recommendation has been shown to carry great weight in patient choice in previous vaccine rollouts. For instance, patients receiving a healthcare provider recommendation for the HPV vaccine were almost five times more likely to receive one dose of the vaccine compared with those who received no physician recommendation [29]. Continued discussions about the value of COVID-19 vaccination and the risks versus benefits are essential [30]. According to Dr. Edje, patient advocates are an integral part of patient care, and advocates discussing the merit of COVID-19 vaccination is a crucial part of patient education [31]. 

Next, the pharmaceutical companies have played a role in vaccine hesitancy given questions about their profiting from a global health emergency. The four ethical principles for all parties engaged in allocating vaccines are optimizing vaccine production, fair distribution, sustainability, and accountability [32]. However, companies are, by definition, profitmaking enterprises, which further complicates the web of the stakeholders. As Abassi puts it, “Politicians and industry are responsible for this opportunistic embezzlement. So too are scientists and health experts. The pandemic has revealed how the medical-political complex can be manipulated in an emergency—a time when it is even more important to safeguard science” [33].

Finally, the last, and perhaps most important, stakeholder in this issue is the patients. Noting that disadvantaged groups in the U.S. often have poor access to the COVID-19 vaccinations, policy reform is essential to increase the number of vaccinated people [34]. These authors also mention that vaccines should be in easily situated and “trusted” locations, yet only 18 jurisdictions use a disadvantage index. While the ethicality of forced vaccine mandates in the general public is still being debated, there remains 22% of the U.S. population that has not received a COVID-19 vaccine dose as of July 2022 [35]. Every effort must be made by hospitals to protect these individuals from COVID-19, regardless of their vaccination choice. Healthcare worker vaccinations would eliminate one of the avenues by which the virus could spread in a hospital setting to this unvaccinated population. 

It remains challenging to convince unvaccinated HCWs to get vaccinated. As already mentioned, it is the “ambassadorial” duty of healthcare professionals to encourage vaccination, despite some HCWs’ misgivings about pharmaceutical companies profiteering from the vaccine development or “manipulation” of clinical trial data. 

## 4. Weighing the Policy Alternatives

This section considers some of the policy alternatives to consider regarding the hesitancy of COVID-19 vaccination in HCWs. There will be four alternatives discussed, and they could all potentially benefit the vulnerable patient populations. 

### 4.1. Alternative 1: Adhering to the Mandate for Healthcare Workers

Even before the pandemic, HCWs were required to be vaccinated against several diseases to take care of vulnerable populations, including children, the elderly, and immunocompromised people. The vaccination requirements of HCWs in the U.S. include diphtheria, pertussis, tetanus, hepatitis B, varicella-zoster, measles, mumps, rubella, and influenza [36]. COVID-19 vaccine hesitancy is attributable to several elements, including the novelty of the mRNA vaccine technology and conspiracy theories [6,7]. Regarding the novelty of the vaccine technology, mRNA vaccines have been studied for decades [37]. However, it has been stability rather than safety that has eluded researchers, as exogenous RNA-based therapeutics are targets for degradation in the body. Recent advances in mRNA packaging have allowed for stable mRNA vaccine development to be scaled up. Secondly, conspiracy theories as outlandish as microchip control and biosurveillance have gained traction [38]. These non-factual statements must be combatted by physicians with patience and evidence-based medicine. 

On 15 March 2022, the Supreme Court announced the mandate for HCWs to be vaccinated against COVID-19 [39]. However, this mandate has increased healthcare worker attrition due to burnout, lifestyle, and other factors. Despite the resistance, we contend that this is the most viable option given the data surrounding the vaccine’s efficacy and the duty of the healthcare community to serve the patients. 

### 4.2. Alternative 2: Not Requiring Healthcare Workers to Be Vaccinated

If HCWs were allowed to remain unvaccinated, this would enable many healthcare professionals who have left the profession to return, potentially alleviating the nationwide shortages of HCWs. Governor Greg Abbott signed an executive order banning the vaccine mandate in Texas. Politicians in Mississippi and Florida had similar opinions regarding the vaccine mandate [40]. Interestingly, despite the laxity in COVID-19 vaccination requirements, the Texas Hospital Association struggles with obtaining enough nurses to help care for patients during surges and beyond [41]. Therefore, this may suggest that there are not enough healthcare professionals to cope with nationwide shortages due to the pandemic. Even accounting for the deaths of HCWs during the pandemic, there is not enough trained workforce to take over the deficits of medical workers in healthcare. 

### 4.3. Alternative 3: Supplying Adequate Personal Protective Equipment (PPE) 

Vaccine mandates for healthcare professionals have led to severe shortages of HCWs in many different capacities. According to data from the Centers for Disease Control and Prevention (CDC), 30% of HCWs were unvaccinated as of September 2021 [10]. In nursing homes, numerous potentially preventable deaths due to the delta variant of COVID-19 were caused by HCWs not having been vaccinated [12]. An alternative strategy would be to ensure healthcare systems could afford to purchase enough PPE for unvaccinated HCWs and patients during the COVID-19 surges so that COVID-19 vaccination would not have to be mandated. However, the cost of PPE must be accounted for, which would further inflate healthcare costs [42]. With new variants on the rise that are resistant to the current vaccines on the market, it is worth considering that PPE mandates, in addition to vaccine mandates, may be necessary to counter the recent surges [43].

### 4.4. Alternative 4: Doing Nothing in the Face of the Pandemic

The costs to the healthcare system of doing nothing would be catastrophic in terms of death rates and financial costs for both patients and HCWs. Arguing that turning a blind eye to COVID-19 surges would result in chaos for healthcare, Aschwanden cautions about the “false promise of herd immunity” [44]. The U.S. has not adopted the herd immunity approach, but countries are still reeling from the consequences. In fact, the constantly evolving variants of COVID-19 and risk of re-infection further discredits a herd immunity approach. An immunologist from the Scripps Institute, Aschwanden, quotes Andersen, and maintains, “Attempting to reach herd immunity via targeted infections is simply ludicrous. In the U.S., probably one to two million people would die. He goes on to say, ‘Vaccination is the only ethical path to herd immunity’” [44].

## 5. Conclusions

The risk of transmission from HCWs to vulnerable populations has prompted the national mandate for COVID-19 vaccination of HCWs, unless there is a specific exemption due to severe anaphylaxis to the first dose of the vaccine or, in certain states, religious exemptions. This mandate is supported by the Department of Health and Human Services or the Centers for Disease Control and Prevention (CDC) authority to enact measures to reduce the transmission or spread of infectious diseases [45].

The Centers for Medicare and Medicaid Services have specified COVID-19 vaccination requirements for healthcare professionals. The Phase I conditions were that facilities must have all policies and procedures for vaccinating and tracking staff in healthcare facilities, including clear guidelines for exemptions. The Phase II requirements are that all staff be vaccinated. If a facility does not meet the criteria, it risks fines and Medicare or Medicaid payment loss. Therefore, it is mandated that healthcare professionals receive the COVID-19 vaccinations to be able to work in most healthcare settings. Considering the precedent by other healthcare vaccine mandates, the efficacy of the vaccine, and lack of viable alternatives, the COVID-19 mandate remains the most efficacious option for protecting both patients and providers. 

## Data Availability

Not applicable.

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
