# Peer review of "COVID-19 Vaccine Mandates for Healthcare Professionals in the United States"

_vaccines, 2022, doi:10.3390/vaccines10091425_

Round 1
Reviewer 1 Report
This is much more of an editorial rather than a careful dissection of the problem and it's root causes. There are some important missing facts. While the author's state that 30% of health care workers are unvaccinated, there is no data presented on how many health care workers actually left health care because of the vaccine mandate. There are many reasons for the great resignation and we cannot assume it is because of the vaccine mandate. Second, there is an assumption made that herd immunity can be achieved through vaccination. In fact, there is no proof that this can be done in the midst of a novel pandemic. Herd immunity has been achieved to a number of viral diseases typically over the course of 5-20 years of vaccine campaigns with only one disease, small pox, being eradicated. The emergence of variants makes a strategy of vaccination unlikely to be confer herd immunity in a short period of time. Vaccination is most protective of the individual who becomes infected preventing hospitalization and death. Vaccination has not, as yet, truly lowered the rate of transmission because of the emergence of variants.
Stylistically, this is a bit roughly written with transitions that are hard to follow. There is no mention of the political aspects of the campaign against vaccine mandates. This is playing an important role in how people decide if they are going to get vaccinated.
Reviewer 2 Report
Thank you for the invitation. I have read this manuscript with great interest. The authors have tried to summarize the current scenario of vaccine mandates in the USA and provided a few solutions as alternatives.
Since this is a small opinion paper, I have no major concern about the suggestions proposed by the authors. However, I would like to suggest considering the literature review on the opinions of healthcare professionals on vaccine mandates across the world. There are few studies on this topic evaluating the perception, attitude, and opinion of the healthcare professionals towards the vaccine mandates, associated facilitators, and barriers. These findings in the form of a table, or text will strengthen the importance of vaccine mandates, which will help to convince the target audience for future administrative moves.
Since this paper is in favor of vaccine mandates, addition of the results underscoring the concerns of the healthcare professionals will help the authorities for educational and counselling programs for professionals.
